# EpoR Activation Stimulates Erythroid Precursor Proliferation by Inducing Phosphorylation of Tyrosine-88 of the CDK-Inhibitor p27^Kip1^

**DOI:** 10.3390/cells12131704

**Published:** 2023-06-23

**Authors:** Fragka Pegka, Nathalie Ben-Califa, Drorit Neumann, Heidelinde Jäkel, Ludger Hengst

**Affiliations:** 1Institute of Medical Biochemistry, Biocenter, Medical University of Innsbruck, 6020 Innsbruck, Austria; 2Department of Cell and Developmental Biology, Sackler Faculty of Medicine, Tel Aviv University, Tel Aviv 69978, Israelhisto6@tauex.tau.ac.il (D.N.)

**Keywords:** p27, EpoR, erythropoiesis, tyrosine phosphorylation, cell cycle

## Abstract

Erythrocyte biogenesis needs to be tightly regulated to secure oxygen transport and control plasma viscosity. The cytokine erythropoietin (Epo) governs erythropoiesis by promoting cell proliferation, differentiation, and survival of erythroid precursor cells. Erythroid differentiation is associated with an accumulation of the cyclin–dependent kinase inhibitor p27^Kip1^, but the regulation and role of p27 during erythroid proliferation remain largely unknown. We observed that p27 can bind to the erythropoietin receptor (EpoR). Activation of EpoR leads to immediate Jak2–dependent p27 phosphorylation of tyrosine residue 88 (Y88). This modification is known to impair its CDK–inhibitory activity and convert the inhibitor into an activator and assembly factor of CDK4,6. To investigate the physiological role of p27–Y88 phosphorylation in erythropoiesis, we analyzed p27^Y88F/Y88F^ knock–in mice, where tyrosine–88 was mutated to phenylalanine. We observed lower red blood cell counts, lower hematocrit levels, and a reduced capacity for colony outgrowth of CFU–Es (colony–forming unit–erythroid), indicating impaired cell proliferation of early erythroid progenitors. Compensatory mechanisms of reduced p27 and increased Epo expression protect from stronger dysregulation of erythropoiesis. These observations suggest that p27–Y88 phosphorylation by EpoR pathway activation plays an important role in the stimulation of erythroid progenitor proliferation during the early stages of erythropoiesis.

## 1. Introduction

Erythropoiesis is a complex process during which hematopoietic stem cells (HSCs) give rise to mature red blood cells (RBCs). It is a precisely regulated multi–step process of proliferation and differentiation. Once cells become committed to the erythroid lineage and differentiate into colony–forming unit–erythroid (CFU–E) cells, erythropoietin (Epo) and the erythropoietin receptor (EpoR) are crucial for cell proliferation and survival, but also terminal differentiation [1,2]. EpoR is a type I cytokine receptor with no intrinsic kinase activity. Its cell surface expression and the Epo–dependent activation of the downstream signaling cascades JAK/STAT5, PI3K, and MAPK depend on the tyrosine kinase Jak2 [3,4].

The cyclin–dependent kinase (CDK) inhibitor p27^Kip1^ (CDKN1B, p27) accumulates during erythroid terminal differentiation (ETD) and promotes cell cycle exit [5,6]. p27 has been shown to interact with central G1–phase Cdks (Cdk2, Cdk4, and Cdk6) during ETD. This led to the finding that p27 inactivates CDK2 kinase activity, leading to retinoblastoma protein (Rb) hypophosphorylation and subsequent G1–phase arrest, cell cycle exit, and terminal differentiation [5]. In addition, lower p27 levels during the final stages of erythropoiesis led to anemic mice with a reduced number of RBCs and an increased number of late–stage erythroblasts [7]. Furthermore, p27 is a direct target of the transcription factor erythroid Krüppel–like factor (EKLF/KLF1), and although late–stage *EKLF^−/−^* erythroid cells fail to enucleate, overexpression of p27 can partially rescue the enucleation and the cell cycle defect observed in these cells [8].

p27 is an intrinsically disordered protein that controls CDK kinase activities during the G1–phase to S–phase transition of the cell cycle, where it contributes to the restriction point control [9]. In its classical role as a CDK inhibitor, p27 binds and inhibits a broad range of CDK/cyclin complexes. However, if phosphorylated on tyrosine residue 88 (Y88), CDK inhibition by p27 is impaired and the CDK “inhibitor” can promote the assembly and activation of the cyclin D/CDK4,6 complexes [10,11]. At the molecular level, a 3_10_ helix of p27 surrounding unphosphorylated Y88 folds into the ATP–binding pocket of the CDK and prevents ATP binding to the kinase. Following the phosphorylation of Y88, the 3_10_ helix of p27 is ejected from the catalytic cleft of the bound CDKs, leading to partial CDK reactivation. Among other substrates, p27–bound CDK2 can now also phosphorylate p27 on threonine 187, which generates a phosphodegron for its SCF^SKP2^ E3 ubiquitin ligase–dependent proteasomal degradation [12,13].

Several receptor and non–receptor tyrosine kinases that phosphorylate p27 on Y88 have been identified, including Jak2, Lyn, and Src [14,15]. These three kinases play central roles in the Epo/EpoR pathway [3,16,17,18]. Activation of EpoR by Epo binding leads to conformational changes of the receptor [19], and Jak2 associated with activated EpoR phosphorylates tyrosine residues of EpoR within its cytoplasmic domain. These become docking sites for proteins containing SH2 domains, such as STAT5. Since p27 is a substrate of Jak2, it was tempting to speculate that p27 might become phosphorylated following EpoR activation. The phosphorylation of p27 might promote cell proliferation induced by the activation of the EpoR pathway.

In this study, we demonstrate that p27 becomes rapidly phosphorylated on Y88 upon Epo stimulation in hematopoietic cells. This Y88 phosphorylation requires the activation of Jak2 but not of Src family kinases. Interestingly, we observed that p27 can directly associate with the EpoR in vitro and co–immunoprecipitate with EpoR. To investigate the physiological role of this interaction and the Y88 phosphorylation, we investigated erythropoiesis in a novel mouse model, where Y88 has been exchanged for phenylalanine (Y88F). Phenylalanine is a precursor for tyrosine biosynthesis. It lacks only one hydroxyl group and cannot be phosphorylated. p27^Y88F/Y88F^ knock–in animals compensate for the robust p27 inhibitory activity by decreasing p27 expression [20]. Yet, we still observed reduced red blood cell counts and hematocrit levels and decreased ex vivo CFU–E colony formation capacity in comparison to the wild–type animals. These observations suggest that phosphorylation of p27 on Y88 plays an important role during the early stages of erythropoiesis and stimulates the proliferation of CFU–E cells. Increased Epo levels in knock–in animals seem to compensate for the defect in erythropoiesis, probably attenuating the reduction in RBC counts and HCT levels.

## 2. Materials and Methods

### 2.1. Cell Culture and Transfection

All cell lines used were cultured at 37 °C in a humidified atmosphere containing 5% CO_2_. UT–7/Epo cells were cultured with Iscove’s modified Dulbecco’s medium supplemented with 10% FCS, 2 mM L–glutamine, 1% penicillin–streptomycin, and 2 U/mL rhEpo (Sigma–Aldrich, St. Louis, MO, USA). HEK–293T cells were cultured in Dulbecco’s modified Eagle’s medium, 10% FCS, and 1% penicillin–streptomycin, transfected with polyethylenimine (PEI) (ratio of DNA to PEI, 1:3) and harvested 48 h after transfection. UT–7/Epo cell line was previously described [21] and available in the lab. HEK–293T cells were provided by Stephan Geley (Division of Molecular Pathophysiology, Medical University of Innsbruck, Innsbruck, Austria).

### 2.2. SDS–PAGE/Immunoblotting and Antibodies

Protein concentration was determined with the Bio-Rad DC protein assay. Protein samples were resolved on 10% or 13% polyacrylamide gels in an SDS running buffer. After incubation with the primary antibody and the corresponding horseradish peroxidase (HRP)–coupled secondary antibody, protein signals were detected by enhanced chemiluminescence (ECL) with a digital image analyzer ImageQuant LAS4000 from GE Healthcare, Chicago, IL, USA or IMAGEQUANT 800. The following antibodies were used: rabbit anti–p27^Kip1^–C19 (Santa Cruz Biotechnology, Dallas, TX, USA), mouse anti–p27^Kip1^–horseradish peroxidase–coupled antibodies (clone 57, BD Biosciences, Franklin Lakes, NJ, USA), mouse anti–pY88–p27^Kip1^ [14], rat anti–EpoR (GM1201, clone: BCO–3H2), rat anti–EpoR (GM1204) [22], mouse anti–Jak2 (clone 691R5, Invitrogen, Carlsbad, CA, USA), rabbit anti–pY1007/1008–Jak2 (Millipore, Burlington, MA, USA), rabbit anti–pY396 Lyn (Labome/Epitomics, Burlingame, CA, USA), rabbit anti–pY694 STAT5 (Cell Signaling, Danvers, MA, USA), mouse anti–Cyclin A (clone E67.1, Santa Cruz Biotechnology, Dallas, TX, USA), mouse anti–Cyclin E (clone HE12, Santa Cruz Biotechnology, Dallas, TX, USA), rabbit anti–Cyclin A polyclonal antibody T310 [23], mouse anti–GAPDH (clone 6C5, Millipore, Burlington, MA, USA), and mouse monoclonal anti–PSTAIR [24].

### 2.3. pY88p27 Immunoprecipitation (IP)

UT–7/Epo cells were lysed in 5 pellet volumes of IP lysis buffer (20 mM Tris–HCl pH 7.5, 150 mM NaCl, 0.5 % NP–40, 1× SIGMAFast Protease Inhibitors, and 1× PhosStop). The lysates were centrifuged at 23,000× *g* for 30 min at 4 °C. For the detection of pY88p27, the clear supernatant was incubated for 10 min at 65 °C, placed immediately on ice for 10 min, and ultracentrifuged at 50,000× *g* for 30 min. The clear protein lysates, containing heat–stable proteins including p27, were incubated with anti–pY88p27 antibodies covalently coupled to protein–A Sepharose beads (Immunosorb A, Medicargo, Uppsala, Sweden) and after three washing steps the eluate was analyzed by SDS–PAGE. Stock solutions of 5 mM Dasatinib (BMS–354825, Hölzel Biotech, Cologne, Germany) and 2 mM JAK Inhibitor I (Calbiochem, San Diego, CA, USA) in DMSO were used.

### 2.4. Co–Immunoprecipitation (Co–IP)

293T cells were co–transfected with Flag–p27, Jak2, and EpoR in different combinations for 48 h. After harvesting, cells were lysed in 5 pellet volumes of Co–IP lysis buffer (50 mM HEPES pH 7.5, 150 mM NaCl,1% Triton, 1 mM EDTA, 1 mM EGTA, 10% Glycerol, 1× SIGMAFast Protease Inhibitors, 1× PhosStop, and orthovanadate) on ice for 30 min. Cell lysates were centrifuged at 23,000× *g* for 30 min at 4 °C and the protein concentration of the clear supernatant was quantified with the Bio–Rad assay kit. Protein lysates were incubated with 15 µL anti–Flag M2 Affinity Gel (Sigma–Aldrich, St. Louis, MO, USA) in a final concentration of 0.5% Triton Co–IP buffer for 90 min on an overhead rotator at 4 °C. After the incubation, the beads were washed 5 times with 1 mL Co–IP lysis buffer containing 0.5% Triton. The eluate was analyzed with SDS–PAGE.

### 2.5. Recombinant Protein Production and Glutathione–S–Transferase (GST) Pull–Down

GST and the recombinant GST–tagged protein GST–Grb2 were expressed in the *E. coli* strain BL21 (D3) with 0.5 mM Isopropyl β–D–1thiogalactopyranoside (IPTG) for 3 h at 37 °C and the GST–EpoR with 0.1 mM IPTG overnight at 18 °C. The bacteria were lysed in GST lysis buffer (50 mM Tris pH 7.5, 150 mM NaCl, 1 % NP–40, 0.1 mM apoprotein, 0.1 mM pepstatin, 1 mM DTT, and 1 mM PMSF) and the GST–tagged proteins were purified by overnight incubation with glutathione Sepharose beads at 4 °C. The beads were washed three times with lysis buffer and stored at 4 °C in 50 mM Tris pH 7.5, 150 mM NaCl, 0.1 mM apoprotein, 0.1 mM pepstatin, and 10% glycerol. Bacterially expressed recombinant p27, His–tagged p27 N–terminus, and His–tagged p27 C–terminus and its mutants p27 C–, p27 CK–, and p27 ∆SH3 were purified by heat treatment since p27 is a heat stable protein [23]. Bacteria were lysed in 50 mM Tris pH 7.2, 150 mM NaCl, 1% NP–40, 1× protease inhibitors, 1 mM DDT, and 1 mM PMSF, and the lysate was centrifuged for 30 min at 25,000× *g* at 4 °C. The supernatant was boiled for 10 min at 95 °C, placed on ice for 10 min, and ultracentrifuged at 50,000× *g* for 30 min.

### 2.6. In Vivo Experiments

All the animal experiments were approved by the Bundesministerium für Wissenschaft, Forschung und Wirtschaft (Austrian Federal Ministry of Science and Research, license number GZ BMWFW–66.011/0033–WF/V/3b/2016). Male wild–type (WT) and p27^Y88F/Y88^ knock–in (KI) C57BL/6J 7–8–week–old mice were used for the in vivo analysis [20]. Hematocrit (HCT) and hemoglobin (HGB) values, and red blood cell (RBC) counts of peripheral blood were measured with a scil Vet ABC Hematology Analyzer. Reticulocyte analysis was performed as described by Lee et al. [25]. Briefly, 1 μL of peripheral blood was stained with Thiazole orange for an hour at room temperature and cells were analyzed on a BD LSR flow cytometer. Circulating plasma Epo levels from 6 male WT and 5 male knock–in mice were measured using ELISA according to the manufacturer’s protocol (Mouse Erythropoietin Quantikine ELISA, R&D SYSTEMS, Minneapolis, MN, USA). For the RBC survival analysis, mice were injected intravenously with 1 mg NHS–biotin (Sigma–Aldrich, St. Louis, MO, USA). An amount of 5 μL of blood was withdrawn from the tail after one hour (set as the 0 time point) and at the indicated time points for the following 7 weeks. The cells were stained with Ter119–PE and streptavidin–APC and analyzed on a BD LSR flow cytometer.

### 2.7. CFU–E Assay

Six male mice per genotype (WT or knock–in) were used for the colony–forming assay in three independent experiments. The mice were sacrificed by CO_2_ asphyxiation and the bone marrow (BM) was isolated from the femurs in sterile conditions. Briefly, 400,000 cells were mixed with methylcellulose (MethoCult M3234, matrix without cytokines, Stem Cell Technologies, Vancouver, WA, USA), 3 U rhEpo/mL, penicillin, and gentamicin, and distributed in duplicates in 3 cm dishes. Colony numbers were determined after 72 h by light microscopy.

### 2.8. Antibody Staining and Fluorescence–Activated Cell Sorting (FACS)

Freshly isolated bone marrow cells were dissolved in phosphate–buffered saline, 2% fetal calf serum, and 2 mM EDTA (PBS/2% FCS/2 mM EDTA) and passed through a 40 μm strainer. Cells were immunostained with PE–conjugated anti–Ter119 (Biolegend, San Diego, CA, USA) and APC–conjugated anti–CD71 (e–Bioscience, San Diego, CA, USA) antibodies for 30 min at 4 °C in the presence of rat serum. DAPI (Sigma–Aldrich, St. Louis, MO, USA) was used to exclude dead cells. Unstained cells and single–color stainings were used as controls. Cells were analyzed on a BD LSR flow cytometer and data were analyzed with the FlowJo 10.7.1 software.

### 2.9. Statistical Analysis

The Mann–Whitney test, Welch’s test, and two–tailed unpaired Student’s *t*–test were carried out when comparing two groups and are specified in each figure legend. The similarity of variances was tested by the *F*–test. Data were analyzed using GraphPad Prism 9.0.1 software (GraphPad Software, San Diego, CA, USA) and are represented as mean ± standard deviation unless otherwise specified. Statistical significance is indicated as * *p* < 0.05, ** *p* < 0.01, and *** *p* < 0.001.

## 3. Results

### 3.1. Epo Stimulation Causes Rapid Phosphorylation of p27 on Y88

Induction of p27 expression during erythroid terminal differentiation contributes to cell cycle arrest and mitotic exit [5,26,27]. However, little is known about the post–translational regulation of p27 upon EpoR signaling. In order to investigate the regulation of p27 upon Epo binding to EpoR, we used the Epo–dependent human erythroleukemic cell line UT–7/Epo [21]. Epo was depleted from the medium for 18 h and then added to a final concertation of 25 U/mL. Cells were harvested at different time points between 5 min and 8 h following Epo stimulation, and the phosphorylation of p27–Y88 was determined. We observed strong and immediate Y88 phosphorylation of p27 following Epo stimulation (Figure 1A). This phosphorylation occurred with a similar kinetic as the established phosphorylation of Jak2 on Y1007/1008 [28]. Following activation, the mature EpoR (upper band) declined, which may reflect degradation [29]. Interestingly, the decline in Y88 phosphorylation of p27 correlated well with the decline in mature EpoR and preceded the dephosphorylation of Jak2 (Figure 1A).

### 3.2. Phosphorylation of p27 upon Epo Stimulation Depends on Jak2 Activation

p27 is a substrate of different tyrosine kinases, including Jak2, Flt3, BCR–Abl, and members of the Src family [14,15,26,27]. In order to identify the tyrosine kinase responsible for the phosphorylation of p27, we treated UT–7/Epo cells with two different tyrosine kinase inhibitors for 30 min prior to Epo stimulation and subsequently determined the p27–Y88 phosphorylation. We used Dasatinib to inhibit Lyn and other Src family kinases [30], and JAK Inhibitor I to inhibit Jak2. To monitor kinase inhibition, we used phosphorylation–specific antibodies against those kinases. As an additional control, we also monitored the phosphorylation status of their known target, Stat5 [16,17], and as expected we observed reduced phosphorylation upon Epo stimulation in the presence of both tyrosine kinase inhibitors. The activity of Jak2 was monitored by the phosphorylation of tyrosine residues Y1007/Y1008, which are trans– and autophosphorylated [28,29], and the activity of Lyn was tracked by the phosphorylation of tyrosine residue 396, which is inhibited in the presence of Dasatinib [30] (Figure 1B). The inhibition of Lyn did not affect p27 tyrosine–88 phosphorylation, whereas Jak2 inhibition prevented the Y88 phosphorylation of p27 (Figure 1B). This observation indicated that activation of Jak2 is essential for the phosphorylation of p27 upon Epo stimulation and that this phosphorylation does not require Src family kinase activation.

### 3.3. p27 Can Bind to EpoR

Since p27 is rapidly and efficiently phosphorylated following Epo stimulation (Figure 1A), we next investigated whether p27 can be found in a complex with EpoR. We co–transfected HEK–293T cells with p27, EpoR, and/or Jak2. These cells were harvested and p27 was immunoprecipitated. As expected and reported previously, endogenous PSTAIRE–type CDKs and cyclin A co–immunoprecipitated with p27 [31,32,33]. In addition, in the presence or absence of transfected Jak2, EpoR co–immunoprecipitated with p27 (Figure 2). This suggested that either p27 might bind to EpoR complexes independently of Jak2 or that endogenous Jak2 might be sufficient to allow co–immunoprecipitation of EpoR with p27. We also tried to co–immunoprecipitate endogenous EpoR from UT–7/Epo cells with endogenous p27 but were not able to reproducibly recover sufficient complex above background signal. As p27 is predominantly localized in the nucleus, the high nuclear p27 fraction may have prevented the detection of the endogenous p27/EpoR complexes.

### 3.4. p27 Directly Interacts with EpoR In Vitro

EpoR can co–immunoprecipitate with p27 (Figure 2). This could be due to the direct binding of both proteins or the involvement of bridging proteins. To investigate if p27 can directly bind to EpoR, we expressed p27 and EpoR domains in bacteria, since it is very unlikely that bacteria express proteins that specifically interact with both human proteins. The cytoplasmic domain of human EpoR was expressed as a GST fusion protein (GST–hIC EpoR) and was isolated with glutathione Sepharose beads. In pull–down experiments, we observed that GST–hIC EpoR co–precipitated p27, indicating that these proteins can directly interact in vitro (Figure 3A).

We further investigated the interaction with EpoR by using three different p27 mutants: p27C–, p27 CK–, and p27 ΔSH3. p27 C–carries two point mutations and cannot interact with CDKs, p27 CK– carries four point mutations and is deficient for interaction with cyclins and CDKs [34], while the p27 ΔSH3 mutant carries mutations in its proline–rich region preventing interactions with proteins through SH3 domains [14]. All three p27 mutants could bind to GST–hIC EpoR, suggesting that the EpoR/p27 interaction is independent of the cyclin/CDK binding ability or SH3 domain binding ability of p27 (Figure 3A).

To further specify the interaction, we tested the N–terminal CDK–inhibitory domain (amino acids 1–96) of p27 and its C–terminal domain (amino acids 97–198) in the same experimental setting. We used GST–Grb2 as a positive control since it binds tightly to the proline–rich domain of p27 [14,35,36]. We observed that GST–hIC EpoR co–precipitates only the N–terminal domain of p27, while the C–terminal domain failed to interact with the GST–hIC EpoR (Figure 3B).

In order to identify domains in the cytoplasmic domain of EpoR that are required for stable p27 binding, we truncated the intracellular domain of EpoR and fused the fragments to GST. One truncated fragment includes the N–terminal part of the intracellular domain of EpoR (amino acids 273–433). This domain includes the box1 motif, which is required for Jak2 interaction and activation. The second fragment contains the 75 C–terminal amino acids of EpoR (amino acids 434–508). This C–terminal domain includes a cytoplasmic motif called the immunoreceptor tyrosine–based inhibitor motif (ITIM), which is involved in intracellular signaling and can bind, after tyrosine phosphorylation, to the SH2 domains of phosphatases. In addition, the C–terminal domain also includes sequences required for high–affinity SOCS3 binding, STAT1/STAT3 activation [37] or CrkL binding [38]. We incubated these fragments with p27 (Figure 3C) or with N–terminal p27 (unpublished data). While the complete cytoplasmic domain of EpoR (hIC–EpoR) and the N–terminal truncated cytoplasmic domain (amino acids 273–433, hIC–EpoR N) of EpoR could bind to p27, the C–terminal fragment (hIC–EpoR C) failed to bind to the N-terminal fragment of p27 (unpublished data) or full-length p27 (Figure 3C).

**Figure 3 cells-12-01704-f003:**
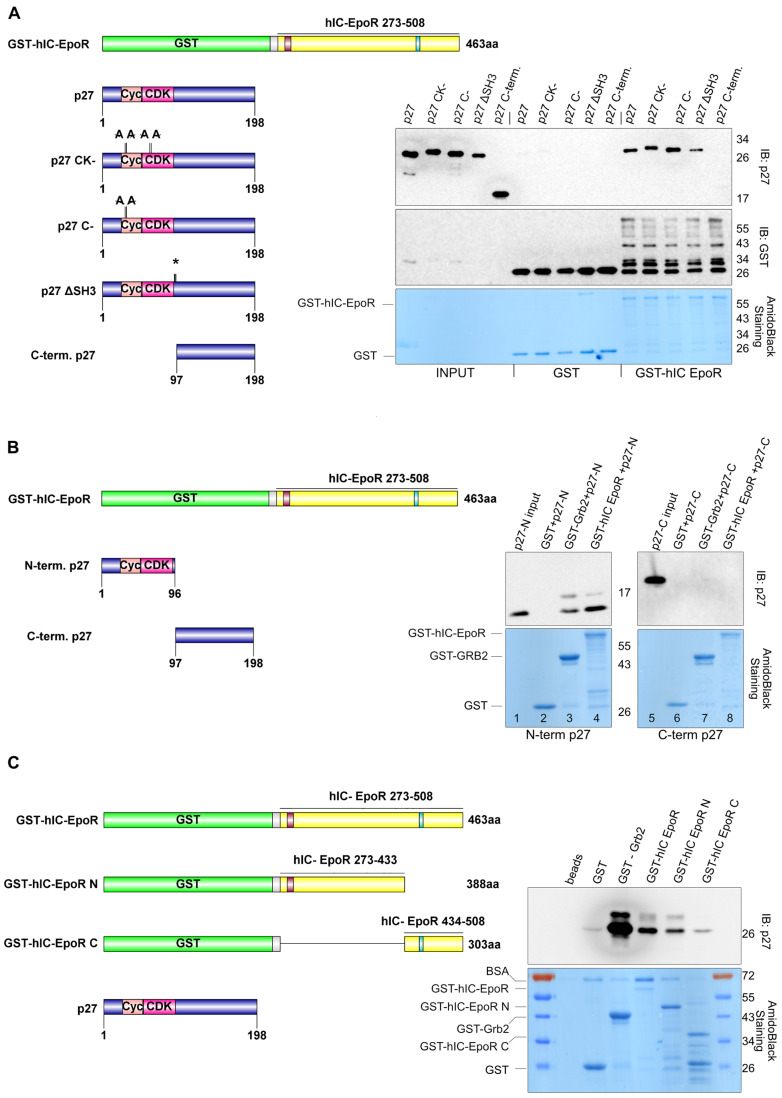
The N–terminal CDK/cyclin binding domain of p27 binds the central cytoplasmic domain of EpoR. (**A**) p27 can directly bind to the intracellular domain of EpoR in vitro. GST, GST–Grb2 as a control, and GST–hIC EpoR recombinant proteins were expressed in *E. coli* BL21 and purified with glutathione Sepharose beads. The intracellular domain of EpoR contains the box1 motif (indicated in red) and the C–terminal ITIM motif (indicated in light blue). The GST or GST–fusion proteins bound on glutathione Sepharose beads were incubated for 2 h at 4 °C with purified p27 or purified p27 mutants. p27 CK– carries four point mutations, namely R30A, L32A, F62A, and F64A, and cannot bind to cyclins and CDKs. p27 C–carries two point mutations (R30A and L32A) and is devoid of cyclin binding, while the ΔSH3 mutant carries point mutations at amino acids 94, 95, and 96 (PPK to GCA, indicated by an *). After extensive washing, the proteins were eluted from the beads by boiling with Lämmli buffer, and GST–hIC EpoR–bound p27 was detected by WB analysis. The GST and GST fusion proteins were detected by amido black staining (bottom panel). (**B**) The N–terminus of p27 (amino acids 1–96) binds to EpoR in vitro. The experimental procedure was performed as described above. The recombinant p27 N–terminus (amino acids 1–96) and C–terminus (amino acids 97–198) were incubated for 2 h at 4 °C with GST (negative control), GST–Grb2 (positive control), and GST–hIC EpoR immobilized on glutathione Sepharose beads. A specific interaction of the p27 N–terminus with the intracellular domain of EpoR was detected by Western blot. In lanes 1 and 5 (input), only the N–terminal or C–terminal p27, respectively, were loaded. (**C**) p27 binds to a central region of the cytoplasmic domain of EpoR (between aa 273–433). Recombinant GST–hIC EpoR (273–508aa) and the deletion constructs GST–hIC EpoR N (273–433aa) and GST–hIC EpoR C (434–508aa) were incubated with full–length p27. Beads and GST were used as negative controls while GST–Grb2 served as positive control. The interaction of p27 with GST–hIC EpoR and GST–hIC EpoR N was detected by immunoblot (upper panel). The amido black stained membrane is shown in the lower panel. The illustrations of the proteins and their domains were generated using IBS version 1.0. [39].

### 3.5. p27^Y88F/Y88F^ Knock–In Mice Are Characterized by Reduced Red Blood Cell Counts and Lower Hematocrit Levels

To investigate the physiological relevance of the p27–Y88 phosphorylation upon EpoR activation, we investigated hematopoiesis in a p27^Y88F/Y88F^ knock–in mouse model recently established in our lab [20]. In these mice, amino acid Y88 of p27 was mutated to the closely related phenylalanine (Y88F) (Figure 4A) [20]. To explore the potential effect of p27–Y88 phosphorylation on erythropoiesis, we compared peripheral blood or bone marrow cells of 7–8-week-old p27 WT or p27–Y88F male mice. As discussed below, enhanced Epo production may compensate for proliferative defects in the red blood cell lineage in these animals. However, the RBCs and the hematocrit (HCT) of the p27–Y88F mice were significantly reduced compared to WT animals (*p* = 0.04118 and *p* = 0.0127, respectively; unpaired *t*–test) (Figure 4B). Thiazole orange staining of peripheral blood cells did not reveal a difference in the percentage of the reticulocytes between WT and knock–in animals (Figure 4C). Even though overall slightly reduced, no statistically significant difference was detected in the hemoglobin (HGB) levels between WT and Y88F mice, possibly due to individual variations in the mice investigated (Figure 4B) or increased compensatory hemoglobin synthesis.

Phosphorylation of p27 on Y88 can lead to its SCF^Skp2^–dependent proteasomal degradation [40]. We therefore expected an increased p27–Y88F protein level. Surprisingly, when we compared p27 levels in the spleen of 7–8-week-old WT and knock–in mice, we observed that the knock–in mice expressed lower levels of the protein (Figure 4D). Similarly, we recently described reduced levels of p27–Y88F in the bone marrow, spleen, and thymus of 10–week-old knock-in animals [20]. The reduced protein level appears not to be caused by transcriptional mechanisms since the mRNA level remains unchanged [20]. The EpoR levels were not significantly altered in knock–in mice (Figure 4D).

### 3.6. Impaired CFU–E Colony–Forming Capacity of BM Cells from p27^Y88F/Y88F^ Mice

p27 phosphorylation of Y88 can convert p27 from an efficient CDK inhibitor to an assembly factor and activator of CDK4,6 and impairs its ability to inhibit CDKs [9,10,14]. To further investigate the potential physiological role of p27–Y88 phosphorylation in controlling cell proliferation during erythropoiesis, we analyzed different stages during this process. Initially, we performed a colony–forming unit–erythroid (CFU–E) assay with cells from WT and knock–in littermates. The CFU–Es are rare progenitors in the bone marrow that generate colonies within 48–72 h in the presence of Epo. Bone marrow from 7–week–old male animals was isolated and cultured in growth–factor–free methylcellulose media in the presence of 3 U/mL Epo. We observed a significant reduction in the number of colonies formed from the bone marrow cells of Y88F knock–in mice compared to WT cells (*n* = 6, *p* = 0.0087, Mann–Whitney U test), indicating a severe defect in the early stages of erythropoiesis and cell proliferation (Figure 5A). This supports the hypothesis that p27–Y88 phosphorylation in response to Epo stimulates progenitor proliferation and that the inability to phosphorylate this residue leads to decreased proliferation when stimulated by Epo. As discussed below, feedback regulation due to increased Epo levels and reduced p27 expression may compensate for this proliferative defect in knock–in animals.

### 3.7. p27^Y88F/Y88F^ Mice Have Fewer Early Basophilic Erythroblasts in the BM

To investigate if there are changes in the later stages of erythropoiesis, the bone marrow of WT and p27–Y88F knock–in mice was isolated and stained for Ter119 and transferrin receptor (CD71). Ter119 is an erythroid–specific antigen that is highly expressed in all erythroblasts after the proerythroblast stage, while CD71 is highly expressed by erythroblasts and early basophilic erythroblasts and declines through erythroid terminal differentiation. Single viable DAPI^−^ Ter119+ cells were further analyzed according to their size (forward scatter area, FSC–A) and CD71 expression, and divided into the following subpopulations: EryA (Ter119^+^CD71^+^FCS–A^high^) consisting of early basophilic erythroblasts, EryB (Ter119^+^CD71^+^FCS–A^low^) representing late basophilic and polychromatic erythroblasts, and EryC (Ter119^+^CD71^−^FCS–A^low^) consisting of orthochromatic erythroblasts (Figure 5B). Although no differences were observed in the total percentage of Ter119^+^ cells (Figure 5C), the percentage of early basophilic progenitors in the bone marrow of Y88F knock–in mice was significantly lower compared to the WT mice (WT *n* = 11, Y88F *n* = 9; *p* = 0.0002, unpaired *t*–test) (Figure 5D). This decrease in early basophilic progenitor cells was consistent with our observation that knock–in mice showed a lower capacity to form CFU–E colonies (Figure 5A). No major differences were detected during the later stages of differentiation. In late erythropoiesis, the robust CDK inhibition by p27–Y88F might facilitate cell cycle exit and differentiation of erythroblasts.

### 3.8. Epo Plasma Levels Are Increased in p27^Y88F/Y88F^ Knock–In Mice

In the CFU–E colony–forming assay we observed a striking difference in the number of colonies between the WT and the knock–in mice. However, when we analyzed the early basophilic erythroblasts and the later differentiation stages of erythropoiesis in vivo, the difference was small. We speculated that the knock–in mice might compensate for the reduced proliferation and erythropoiesis due to the O_2_–mediated feedback regulation of Epo expression. Since the Epo levels are increased in hypoxia, we decided to determine the concentration of Epo in the plasma of WT and knock–in animals. Epo levels were analyzed by ELISA. We observed that the plasma Epo levels of the knock–in mice were increased by 16% on average compared to the WT mice (Figure 5E). These data are consistent with the finding that p27–Y88F knock–in mice compensate for the defect in erythropoiesis in part by increasing Epo expression.

### 3.9. The Lifespan of Erythrocytes in p27^Y88F/Y88F^ Mice Is Slightly Increased

Under steady–state conditions, there is a balance between RBC production in the bone marrow and splenic RBC destruction. To investigate if knock–in mice achieve a similar RBC count compared to WT animals by altered survival, we performed an in vivo RBC survival assay. Mice were intravenously injected with NHS–biotin and blood was withdrawn for the following 7 weeks. The blood cells were stained with Ter119–PE antibodies and streptavidin–APC in order to detect the remaining biotinylated red blood cells. As shown in Figure 5F, the survival in p27–Y88F knock–in animals was slightly higher than in their WT littermates. It is conceivable that the elevated Epo levels contribute to the prolonged survival of erythrocytes, as clinical studies in humans have suggested that Epo might act as a survival factor for red blood cells and could prevent eryptosis [41,42,43].

## 4. Discussion

The phosphorylation status of tyrosine residue 88 (Y88) of p27 can determine its function as a CDK inhibitor or activator of CDK4,6 and control the stability of p27 [9,10,11]. In its classical role, the unphosphorylated p27 protein is a strong CDK inhibitor and haploinsufficient for tumor suppression [44]. Phosphorylation of tyrosine–88 converts the inhibitor into an activator and assembly factor of active cyclin D/CDK4,6 complexes [32,45]. A number of tyrosine kinases have been shown to phosphorylate the Y88 of p27 and inactivate the CDK inhibitor in vitro. We report here that p27 can interact with the erythropoietin receptor (EpoR) and that activation by erythropoietin (Epo) leads to the Y88 phosphorylation of p27. Using a novel mouse model, where tyrosine–88 was mutated to phenylalanine, we observed an impaired ability of CFU–E cells to form colonies. This is consistent with the hypothesis that Epo–induced Y88 phosphorylation of p27 stimulates cell proliferation in the early stages of erythropoiesis. p27–Y88F knock–in animals develop several compensatory mechanisms to counterbalance this deficiency in cell proliferation, including reduced expression of p27–Y88F and increased Epo expression.

We observed that p27 can bind to EpoR. The domains required for this interaction could be identified in the region of amino acids 273–433 of the cytoplasmic domain of EpoR and the N–terminal CDK inhibitory domain of p27 (amino acids 1–96). The p27–EpoR interaction is probably direct, as we used bacterial recombinant proteins to map the interaction domains. Under these conditions, Y88–phosphorylated p27 can still associate with EpoR (unpublished data), suggesting that Y88 phosphorylation does not lead to a dissociation of p27 from EpoR. We could confirm the EpoR–p27 interaction using co–transfection experiments but failed to reproducibly recover complexes of endogenous EpoR with p27. p27 localizes predominantly to the nucleus but shuttles readily to the cytoplasm [46]. The abundant nuclear p27 may have caused a strong background in IPs that prevented us from detecting the endogenous complex. However, when stimulating cells with Epo, we observed fast and Jak2–dependent phosphorylation of p27 on Y88. This p27 phosphorylation declined as the mature EpoR level declined. Interestingly, the reduction in p27–Y88 phosphorylation preceded the decline in Jak2 tyrosine phosphorylation.

To investigate the physiological role of the EpoR–p27 interaction and p27–Y88 phosphorylation in erythropoiesis, we analyzed different stages of erythroid development in a p27^Y88F/Y88F^ knock–in mouse model that was recently developed in our lab [20]. While phosphorylation on Y88 reduces CDK inhibition by p27 and supports the assembly of active CDK4,6/cyclin D complexes and SCF^Skp2^–mediated proteasomal degradation of p27 [11,13,26,47], the mutant p27–Y88F is expected to be a robust CDK inhibitor that cannot be regulated by tyrosine phosphorylation [20]. We therefore speculated that the p27–Y88F protein might impede Epo–induced cell proliferation due to its resistance to tyrosine–kinase–induced inactivation and subsequent proteasomal degradation [13,48]. We measured Epo–dependent proliferation and differentiation using bone marrow cells in CFU–E assays. Cells from knock–in mice showed a reduced capacity to form colonies, which is in agreement with inhibited cell proliferation. We next investigated erythroid terminal differentiation by flow cytometry analysis of bone marrow cells. Consistent with the CFU–E assay, the knock–in mice had a reduced number of early basophilic erythroblasts (EryA). However, during subsequent steps in erythroid terminal differentiation, similar numbers of polychromatic and orthochromatic erythroblasts were observed. Interestingly, a similar phenotype (reduced early basophilic erythroblasts, but a normal number of later erythroblasts) was observed in a Cul4A–haploinsufficient (Cul4A^+/−^) mouse model. It has been suggested that Cul4A is a subunit of a p27–targeting E3 ubiquitin ligase during the early stages of erythropoiesis [6]. Cul4A^+/−^ animals express increased p27 protein. Similar to p27–Y88F animals, where p27 cannot be inactivated by tyrosine phosphorylation, Cul4A^+/−^ mice have fewer erythroid–committed progenitors [6]. This is consistent with the central role of Epo–induced p27–Y88 phosphorylation in the early stages of erythropoiesis.

While the CFU–E assay indicated a strong reduction in the proliferative capacity of p27–Y88F erythroid progenitors, the number of red blood cells in the peripheral blood and the hematocrit were only moderately reduced in p27–Y88F knock–in mice. In comparison to wild–type animals, hematocrit (HCT) levels and RBC counts were decreased by 3.17% (HCT) or 2.95% (RBC) on average. These findings were confirmed in a group of mature mice (22–24 weeks old, Appendix A), where the number of peripheral red blood cells and the hematocrit were also statistically significant but moderately reduced. In both age groups, the parameters hemoglobin, MCV, MCH, and RDW were not significantly altered. The weight of the spleen (Appendix A) as well as extramedullary hematopoiesis in the spleen or liver (unpublished data) were unchanged in the knock–in animals. While the hematocrit and red blood cells were reduced, this reduction was not adequately reflected in the hemoglobin levels and only a tendency of reduced hemoglobin was observed. We speculate that the increased Epo levels stimulate heme synthesis through the activation of PKA [49], and in turn the increased heme induces globin synthesis [50,51,52,53], leading to slightly increased hemoglobin levels in p27–Y88F knock–in mice.

The moderate effect of the p27–Y88F mutation on HCT and RBC numbers is likely caused by several compensatory mechanisms. p27–Y88F knock–in mice express reduced p27–Y88F protein levels in the spleen and in the bone marrow (Figure 4G) [20]. The physiological functions of p27 depend on its levels since p27 is a stoichiometric inhibitor of cyclin–dependent kinases. This is underscored by the observation that p27 is haploinsufficient for tumor suppression [44] and the observation that reduced p27 levels are frequently observed in human tumors [40]. The reduced p27–Y88F level in knock–in animals may allow the proliferation of cells despite their inability to inactivate and destabilize p27 by tyrosine phosphorylation [14].

A second mechanism that likely compensates for the impaired erythroid progenitor proliferation in p27–Y88F mice is the elevated expression of Epo. Erythropoiesis is a tightly regulated process, and its homeostatic regulation involves various factors such as hypoxia, stress, or iron homeostasis. It is strongly regulated by transcription factors, including HIF1a [54]. Epo serves as a central cytokine and promotes erythropoiesis via a positive feedback mechanism [54,55]. Furthermore, Epo plays a protective role by regulating genes associated with apoptosis through pathways such as Jak–Stat and PI3K–Akt signaling [56,57,58,59]. It is likely that the reduced oxygen supply due to the decreased number of red blood cells in p27^Y88F/Y88F^ knock–in animals causes mild hypoxia that activates transcription factors such as HIF1, that in turn lead to increased Epo expression [54,55]. In addition to stimulating progenitor proliferation, Epo also promotes the survival of erythroid progenitors in a steady state or under stress conditions [60,61,62]. The ex vivo CFU–E assays were performed under saturating Epo concentrations. In these assays, we observed a significant reduction in CFU–E colony numbers derived from the mononuclear bone marrow cells of the knock–in mice compared to their wild–type littermates. This indicates impaired proliferation in p27^Y88F/Y88F^ knock–in mice. In vivo, we detected a statistically significant reduction in the number of early basophilic erythroblasts. However, this decrease was moderate. The higher endogenous Epo level in the knock–in mice may also protect the proerythroblasts from undergoing apoptosis.

## 5. Conclusions

A role for p27 in erythropoiesis is well established during terminal differentiation due to a gradual increase in p27 expression [5,6]. Here, we describe a novel interaction between EpoR and p27, leading to the Y88 phosphorylation of the CDK inhibitor. The finding that p27 plays an additional central role in the control of the proliferation of early progenitors is supported by observations in p27 knock–in animals, where an inability to phosphorylate the Y88 of p27 leads to impaired proliferation of early erythroid progenitors. Furthermore, compensatory mechanisms such as higher expression of Epo and decreased expression of p27–Y88F ensure that the decrease in red blood cell counts and hematocrit remains at moderate levels.

## Figures and Tables

**Figure 1 cells-12-01704-f001:**
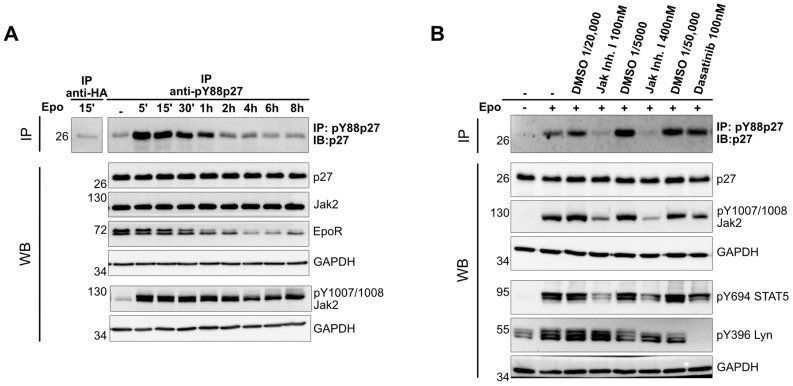
(**A**) p27 becomes phosphorylated on tyrosine 88 (pY88p27) upon Epo stimulation. After Epo starvation for 18 h, UT–7/Epo cells were stimulated with 25 U/mL Epo, incubated, and harvested at the indicated time points. pY88–p27 was detected by immunoprecipitation and Western blot. Anti–HA antibodies were used for the control IP (1st lane). p27, Jak2, and EpoR were detected by WB analysis and GAPDH served as a loading control. (**B**) Y88 phosphorylation of p27 depends on the activation of Jak2. Epo–starved UT–7/Epo cells were treated with the indicated tyrosine kinase inhibitors or the corresponding amount of solvent (DMSO) for 30 min prior to Epo (10 U/mL) stimulation for an additional 15 min. Western blot analysis with the indicated antibodies and pY88–p27 immunoprecipitation/Western blot were performed as described in (**A**). Anti–pY1007/1008 Jak2 and anti–pY396 Lyn antibodies were used to monitor inhibitor treatment, and GAPDH was used as a loading control. Representative images of three independent experiments are shown.

**Figure 2 cells-12-01704-f002:**
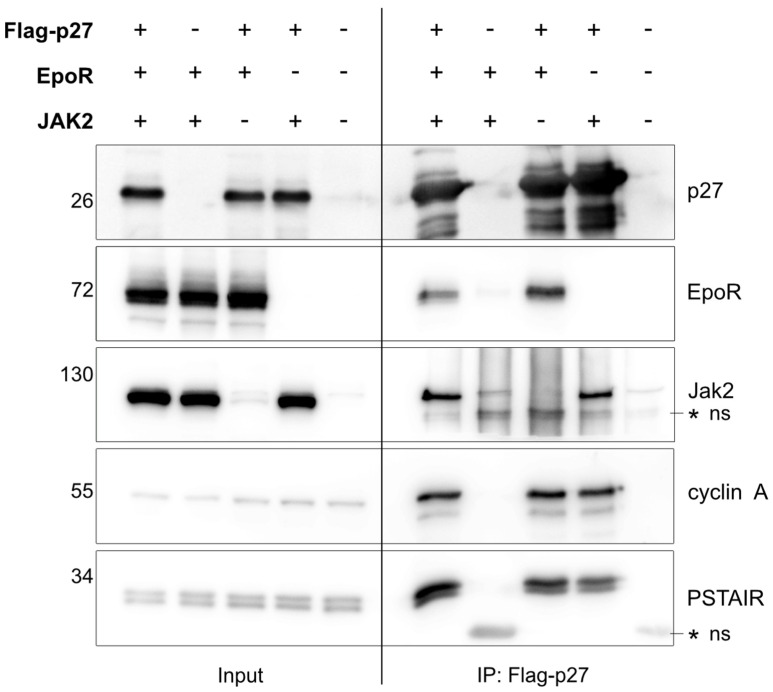
p27 interacts with EpoR in cells. Flag–tagged p27 and/or HA–tagged EpoR and/or untagged Jak2 were co–transfected in HEK–293T cells, as indicated; 48 h after transfection, cells were harvested and subjected to Western blot analysis (**left panel**, “Input”) or immunoprecipitation (IP) of Flag–tagged p27. Co–immunoprecipitated proteins were detected in Western blots (**right panel**, “IP: Flag–p27”). The PSTAIR antibody recognizes the conserved PSTAIRE helix in CDKs such as CDK1 and CDK2, and together with the cyclin A antibody, they were used as positive co–IP controls for the p27. * ns indicates non–specific bands.

**Figure 4 cells-12-01704-f004:**
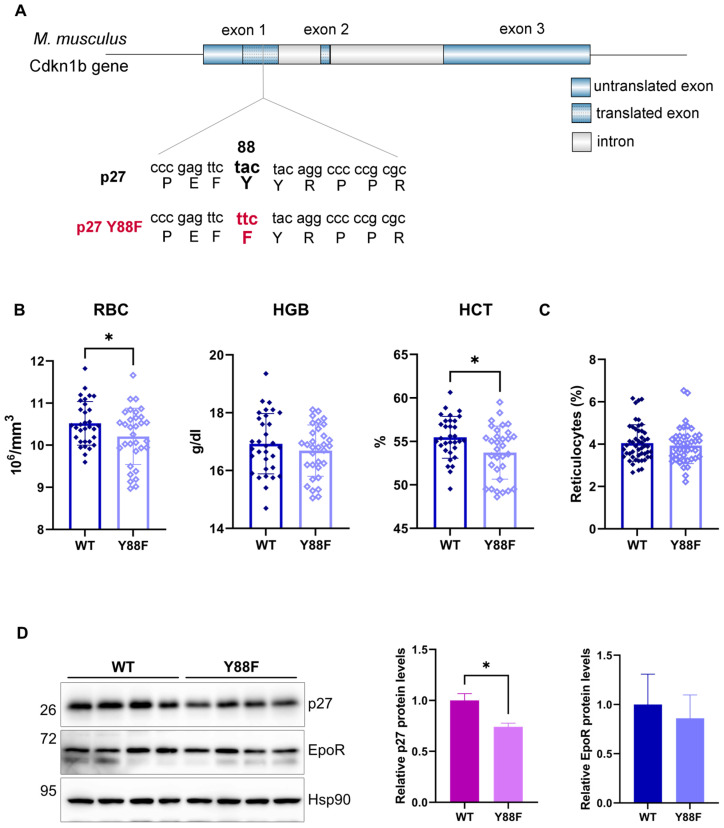
Erythropoiesis in p27–Y88F knock–in animals. (**A**) p27^Y88F/Y88F^ mouse knock–in model. The sequence of p27 was mutated to generate the amino acid exchange tyrosine–88 to phenylalanine as described [20]. (**B**) Comparison of red blood cell parameters uncovers reduced red blood cell count and hematocrit levels in p27–Y88F knock–in mice. Red blood cell (RBC) counts, and hemoglobin (HGB) and hematocrit levels (HCT) of 7–8–week–old male WT and p27^Y88F/Y88F^ mice. Blood values were measured with the scil Vet ABC Analyzer. Levels of significance were calculated using the unpaired *t*–test. WT *n* = 32, KI *n* = 32. (**C**) Flow cytometry analysis of reticulocytes in peripheral blood using Thiazole orange staining. (**D**) Analysis of EpoR and p27 protein levels in the spleen of 7–week–old WT and p27–Y88F mice. Protein levels of EpoR, p27, and Hsp90 as a loading control are shown. The right panels show the quantification of EpoR (upper band) or p27 proteins, normalized to Hsp90. EpoR or p27 levels from WT spleens were set to 1. The Mann–Whitney test was used for statistical analysis, where * *p* = 0.0286. (WT *n* = 4, KI *n* = 4).

**Figure 5 cells-12-01704-f005:**
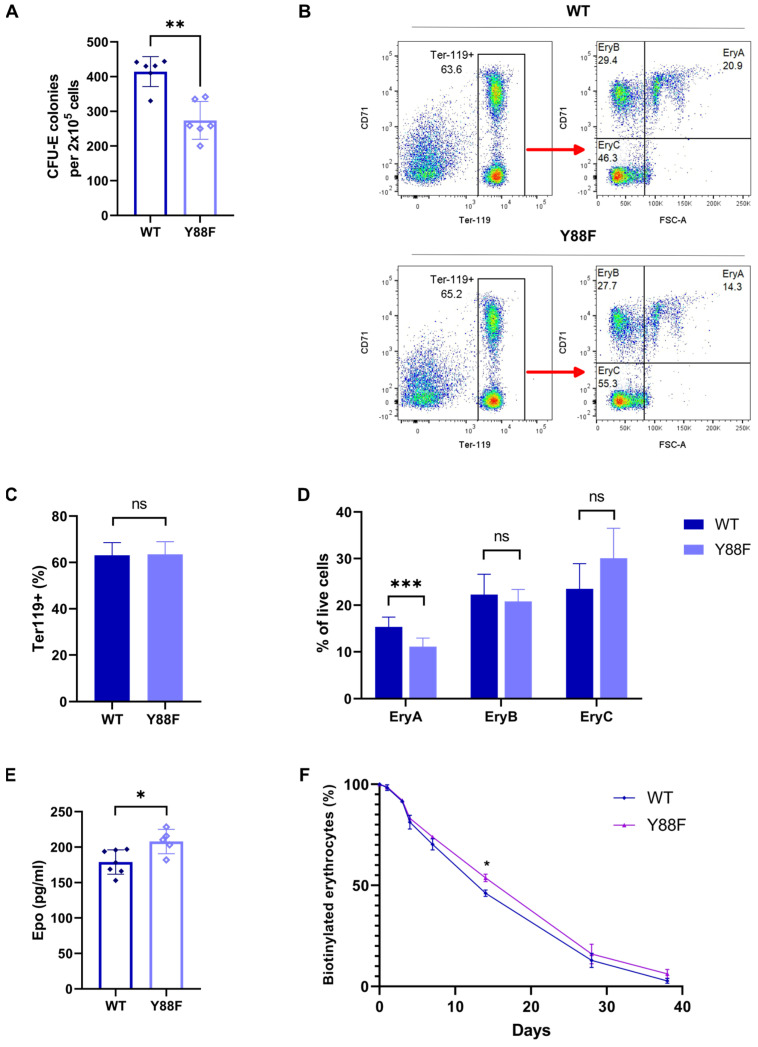
(**A**) Reduced CFU–E colony formation of p27–Y88F bone marrow cells. CFU–E colony formation assay of WT and p27–Y88F bone marrow cells in growth–factor-free methylcellulose containing 3 U/mL rhEpo (*n* = 6 males per genotype, 49–50 days of age, two independent experiments). Statistical significance was determined by the Mann–Whitney U test (** *p* = 0.0087). (**B**) Gating strategy for erythroblast analysis by flow cytometry. Bone marrow cells of WT (upper panel) and p27–Y88F (lower panel) mice were labeled with Ter119 and CD71–specific antibodies. Viable single cells (DAPI–negative) were analyzed for Ter119 expression and further subdivided according to CD71 expression and size (FSC–A) (right panel) into 3 groups: EryA (Ter119 high, CD71 high, and FSC high), EryB (Ter119 high, CD71 high, and FSC low) and the most mature EryC (Ter119 high, CD71 low, and FSC low). (**C**) Percentage of Ter119^+^–positive cells in the bone marrow. No difference in the total number of Ter119^+^ erythroblasts between WT and p27^Y88F/Y88F^ mice was detected in the bone marrow. (ns: not significant) (**D**) Percentage of erythroblast populations in the bone marrow. (WT *n* = 11, p27^Y88F/Y88F^
*n* = 9.) Statistical significance was determined by the Mann–Whitney U test (*** *p* = 0.0002) (ns: not significant). (**E**) Increased plasma Epo levels in p27^Y88F/Y88F^ mice. Plasma Epo levels were determined by ELISA. Statistical significance was determined by the Mann–Whitney U test (* *p* = 0.0177) (WT *n* = 7, KI *n* = 5). (**F**) Red blood cell survival. Mice were intravenously injected with NHS–biotin and 5 μL of blood was drawn from the tail vein at the indicated time points. The blood was labeled with APC–conjugated streptavidin and Ter119–PE. Ter119^+^ NHS–biotin^+^ cells at day 0 were set to 100. The percentage of biotinylated RBCs per timepoint was calculated as the ratio of Ter119^+^ NHS–biotin^+^ cells to the Ter119^+^ NHS–biotin^+^ cells at day 0. (WT *n* = 4, KI *n* = 4, * *p* = 0.0286, Mann–Whitney U test).

## Data Availability

All data are contained within the article or Appendix A.

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
