# Peer review of "EpoR Activation Stimulates Erythroid Precursor Proliferation by Inducing Phosphorylation of Tyrosine-88 of the CDK-Inhibitor p27Kip1"

_cells, 2023, doi:10.3390/cells12131704_

Round 1

Reviewer 1 Report

In this study, Pegka and co-workers have investigated the role of p27 Tyr88 phosphorylation in erythropoiesis. They show that erythropoietin (Epo) rapidly stimulates p27 phosphorylation on Tyr88 by a Jak2-dependent and SFK-independent mechanism. They also show that p27 directly interacts with the Epo receptor. To address the physiological role of this regulatory mechanism, they used a p27Y88F/Y88F knock-in (p27 KI) mouse model. They observed that p27 KI mice have normal red blood cell (RBC) counts and hemoglobin levels, but lower hematocrit and decreased ex vivo CFU-E colony formation capacity. Compensatory changes in p27 and Epo expression may explain the mild erythropoiesis phenotype as the authors suggest.

Overall, this is a well written and carefully executed study with generally convincing results. The weakness of this study is that it is short in terms of novelty and impact. The role of p27 Tyr88 phosphorylation in cell cycle progression has been documented in many other studies, including work by this same group. The authors have previously reported the cytokine-dependent Tyr88 phosphorylation of p27 by Jak2. The role of p27 in erythropoiesis has been studied. Furthermore, the erythropoiesis phenotype of p27 KI mice is very mild and doesn’t bring any novel cellular or physiological insight. The compensatory changes observed in p27 and Epo expression in p27 KI mice are interesting, although not unexpected, but the underlying mechanisms are not further investigated. 

Additional comments: 

- to validate their suggestion that Epo-stimulated p27 Tyr88 phosphorylation plays an important role in the proliferation of early erythroid progenitors, the authors could measure erythropoiesis after subjecting the mice to hypoxia or intense exercise. 

- the observation that p27 KI mice show no difference in RBC counts and Hb levels essentially means that there is no erythropoiesis phenotype physiologically. The observed difference in hematocrit in absence of changes in MCV is most likely a statistical issue. This point should be clarified.

- the mapping of the binding site of p27 on EpoR, while carefully done, is somewhat disconnected from the physiological assessment of erythropoiesis. It does not bring much information to the story. 

Reviewer 2 Report

Dear Authors,

Please find my comments below.

Reviewer 3 Report

Cells Manuscript #2356854

Title: EpoR activation stimulates erythroid precursor proliferation by inducing phosphorylation of tyrosine -88 of the CDK inhibitor p27Kip1

Corresponding Author Heidelinde Jakel and Ludger Hengst

The manuscript from Pegka et al. demonstrates the p27 is phosphorylated by Jak2 upon Epo stimulation. p27 binds to the Epo receptor and is phosphorylated in tyrosine 88. The authors show that a knock mutation of p27-Y88F exhibits a mild anemia and defects in the proliferation of basophilic erythroblasts. They also observe a decrease in CFU-E in the mutant mice.

Overall, the authors do an excellent job demonstrating that p27 binds to the EpoR and is phosphorylated by Jak2 on Y-88. The biological experiments are less persuasive because the effects of the KI mutation are modest.

The in vivo analysis was done of 7–8-week-old mice. Mice become full grown and mature at 8-9 weeks. Did the authors look at older mice 10-14 weeks as they would be more indicative of a defect in steady state erythropoiesis. Conversely, during the first 8 weeks after birth, splenic stress erythropoiesis augments bone marrow erythropoiesis as the mice grow. Did you measure erythroid parameters at birth, weaning or at 5-6-weeks of age?    

Have you tested whether the KI mice have a defect in the respond to acute anemia induced by phenylhydrazine injection?

Are the spleens of the KI mice enlarged? Do they have CFU-E in them? You tested CFU-E, but is there a defect in BFU-E?

Figure 5B, you should show the flow diagrams for the mutant and the control.

Round 2

Reviewer 3 Report

The authors have addressed my concerns.

Author Response

Thank you!